Mitogenomics and phylogenetics of twelve species of African Saturniidae (Lepidoptera)

Nethavhani Zwannda 1
Straeuli Rieze 1
Hiscock Kayleigh 1
Veldtman Ruan 2 3
Morton Andrew 4
Oberprieler Rolf G. 5
van Asch Barbara bva@sun.ac.za 1
1 Department of Genetics, University of Stellenbosch , Stellenbosch , Western Cape , South Africa
2 Department of Conservation Ecology and Entomology, University of Stellenbosch , Stellenbosch , Western Cape , South Africa
3 Kirstenbosch Research Centre, South African National Biodiversity Institute , Cape Town , Western Cape , South Africa
4 Unaffiliated , Cape Town , Western Cape , South Africa
5 Australian National Insect Collection, Commonwealth Scientific and Industrial Research Organisation , Canberra , Australia
Gillespie Joseph
Electronic publication date: 2022 Apr 18
Publication date: 2022
Volume: 10
Electronic Location ID: e13275
Received 2022 Jan 14; Accepted 2022 Mar 24
Copyright: ©2022 Nethavhani et al.
Copyright year: 2022
Copyright holder: Nethavhani et al.
License: This is an open access article distributed under the terms of the Creative Commons Attribution License, which permits unrestricted use, distribution, reproduction and adaptation in any medium and for any purpose provided that it is properly attributed. For attribution, the original author(s), title, publication source (PeerJ) and either DOI or URL of the article must be cited.
License URL: https://creativecommons.org/licenses/by/4.0/

Keywords: Phylogenetics, Eochroini, Attacini, Edible insects, Saturniidae, Southern Africa, Micragonini, Bunaeini, Mitogenome

Funding: The National Research Foundation of South Africa 121293 The South African Department of Forestry, Fisheries and the Environment (DFFtE) The Foundational Biodiversity Information Programme 128333 This study was funded by the National Research Foundation of South Africa (Grant No. 121293). Ruan Veldtman was funded by The South African Department of Forestry, Fisheries and the Environment (DFFtE). Zwannda Nethavhani received a PhD bursary from the Foundational Biodiversity Information Programme (Grant No. 128333). The funders had no role in study design, data collection and analysis, decision to publish, or preparation of the manuscript.

==============================
African Saturniidae (Lepidoptera) include numerous species consumed at the caterpillar stage throughout the continent, and their importance to local communities as a source of nutrition and seasonal income cannot be overestimated. However, baseline genetic data with utility for the characterization of their diversity, phylogeography and phylogenetic relationships have remained scarce compared to their Asian counterparts. To bridge this gap, we sequenced the mitochondrial genomes of 12 species found in southern Africa for comparative mitogenomics and phylogenetic reconstruction of the family, including the first representatives of the tribes Eochroini and Micragonini. Mitochondrial gene content and organization were conserved across all Saturniidae included in the analyses. The phylogenetic positions of the 12 species were assessed in the context of publicly available mitogenomes using Bayesian inference and maximum likelihood (ML) methods. The monophyly of the tribes Saturniini, Attacini, Bunaeini and Micragonini, the sister relationship between Saturniini and Attacini, and the placement of Eochroa trimenii and Rhodinia fugax in the tribes Eochroini and Attacini, respectively, were strongly supported. These results contribute to significantly expanding genetic data available for African Saturniidae and allow for the development of new mitochondrial markers in future studies.

Introduction

The family Saturniidae (Lepidoptera) comprises approximately 3,400 species in 180 genera distributed in a diversity of terrestrial habitats worldwide (Kitching et al., 2018). The African fauna of the family comprises about 400 species in 50 genera, classified in six tribes (Oberprieler, Morton & Noort, 2021). The pupal cocoons of several Asian saturniids are used for silk production, e.g., Antheraea assamensis, Antheraea pernyi, Eriogyna pyretorum, Saturnia japonica and Samia cynthia ricini (Jiang et al., 2009; Kim et al., 2012; Singh et al., 2017; Kim et al., 2018; Liu et al., 2021). Sericulture is not generally practiced in Africa and the potential of African silk-producing species such as Argema mimosae (Saturniini) and Epiphora bauhiniae (Attacini) is largely untapped, although traditional uses are documented (Peigler & Oberprieler, 2017). Nevertheless, the larvae of at least 50 saturniid species are consumed in southern, central and western Africa, where they represent important sources of nutrition and income to rural communities (Jongema, 2015; Kelemu et al., 2015; Malaisse, Mabossy-Mobouna & Latham, 2017; Hlongwane, Slotow & Munyai, 2020; Sekonya, McClure & Wynberg, 2020; Hlongwane, Slotow & Munyai, 2021; Kusia et al., 2021). However, large-scale harvesting of caterpillars from the wild is a threat to populations (McGeoch, 2002), exacerbated by lack of distribution data and monitoring.

In southern Africa, the most popular edible caterpillars are those of Gonimbrasia belina and Gynanisa maja, commonly known as mopane worms. Other edible Saturniidae recorded in southern Africa include species in the tribes Bunaeini (genera Athletes, Bunaea, Bunaeoides, Bunaeopsis, Cinabra, Cirina, Gonimbrasia, Gynanisa, Heniocha, Imbrasia, Lobobunaea, Melanocera, Nudaurelia, Pseudobunaea and Rohaniella), Micragonini (Goodia and Micragone) and Urotini (Pseudaphelia, Urota and Usta) (Silow, 1976; Nonaka, 1996; Thomas, 2013; Jongema, 2015; Malaisse, Mabossy-Mobouna & Latham, 2017; Hlongwane, Slotow & Munyai, 2021). However, knowledge regarding the exact number of species is limited due to taxonomic instability and inconsistencies in the family Saturniidae (Kitching et al., 2018; Siozios et al., 2020). Therefore, specimens belonging to different species are often assigned the same species name and vice versa (Siozios et al., 2020). As a result, the total number of Saturniidae species found and consumed in Africa may be either under- or over-reported, with implications for implementing long-term conservation measures. Despite their importance, no formally protected areas specifically designed for the conservation of these species exist in southern Africa, or for any other edible insect for that matter (Dzerefos, 2018).

Mitogenomes are widely used for phylogenetic reconstruction in insects, and deeper taxonomic coverage and recent methodological developments have allowed for improved recovery of relationships among taxa, even those affected by compositional heterogeneity and accelerated evolutionary rates (Song et al., 2016). Mitochondrial phylogeny estimates of Saturniidae have mostly been limited to Asian species (Jiang et al., 2009; Zhang, Zhao & Zhou, 2016; Kim et al., 2017a; Kim et al., 2017b; Li et al., 2017; Saikia, Nath & Devi, 2019; Kim et al., 2009; Liu et al., 2021), and only two African species in the tribe Bunaeini (G. belina and Gy. maja) were recently included (Langley et al., 2020), representing approximately 1% of the total species recorded in Africa and 4% of all saturniid mitogenomes publicly available. Thus, assessments of phylogenetic relationships, phylogeographic structure and genetic diversity of African Saturniidae have been hampered by limited baseline genetic information.

In this study, we generated mitogenome sequences for 12 African Saturniidae species in four tribes and assessed their phylogenetic position in the family: Eochroa trimenii (Eochroini), Bunaea alcinoe, Heniocha apollonia, Heniocha dyops, Gonimbrasia tyrrhea, Nudaurelia cytherea, Nudaurelia wahlbergii (Bunaeini), Epiphora bauhiniae (Attacini), Ludia delegorguei, Holocerina smilax, Vegetia ducalis and Vegetia grimmia (Micragonini). This novel dataset covers 24% of the total number of species occurring in South Africa, Botswana and Namibia (R. Oberprieler, pers. obs., 2022) and is a significant improvement on the present paucity of mitogenomic data for sub-Saharan Saturniidae.

Materials & Methods

Specimen collection, morphological identification, DNA extraction

Adult and caterpillar specimens were collected at several locations in South Africa and Namibia, between January and March 2020 (Fig. 1, Table 1) under permits approved by the Ezemvelo KZN Permits Office (OP408/2020), Cape Nature (CN44-59-13497) and National Commission on Research Science and Technology (AN20190911). Specimens were euthanized by freezing within a few hours of collection, whenever field conditions allowed. Species identification of adults and caterpillars was made by R. Oberprieler based on photographs of reference specimens using current literature (Pinhey, 1972; Oberprieler, 1995; Staude et al., 2016; Staude et al., 2020). Legs from each adult and caterpillar were excised and stored in 100% ethanol at −20 °C until DNA extraction. Total DNA was extracted from one leg from each adult and caterpillar using a standard phenol-chloroform protocol (Sambrook & Russell, 2012).

Figure 1 Representative specimens of 12 African Saturniidae species.

Larva and adult specimens used in this study for new mitogenomic sequencing and phylogenetic analyses or collected from the same area. (A) Nudaurelia wahlbergii (Boisduval, 1847), (B) Nudaurelia cytherea (Fabricius, 1775), (C) Bunaea alcinoe (Stroll, 1780), (D) Gonimbrasia tyrrhea (Cramer, 1776), (E) Heniocha apollonia (Cramer, 1779), (F) Heniocha dyops (Maassen, 1872), (G) Vegetia ducalis (Jordan, 1922), (H) Vegetia grimmia (Geyer, 1832), (C) Holocerina smilax (Westwood, 1849), (D) Ludia delegorguei (Boisduval, 1847), (J) Eochroa trimenii (C. & R. Felder, 1874), and (L) Epiphora bauhiniae (Guérin-Méneville, 1832).

DNA barcoding

The standard COI barcoding region was amplified using the primer pair LepF/LepR in a total reaction volume of 5 µL containing 2.5 µL of Qiagen Multiplex PCR Kit (QIAGEN), 0.5 µL of each primer, 0.5 µL of Milli-Q water and 1 µL of template DNA. PCR amplifications were performed with initial denaturation at 95 °C for 15 min; 35 cycles of 94 °C for 30 s, 50 °C for 90 s; followed by a final extension at 72 °C for 10 min. Sequencing reactions were performed using LepR with the BigDye Terminator v3.1 Cycle Sequencing Kit (Applied Biosystems, Waltham, MA, USA), at the Central Analytical Facilities of Stellenbosch University. The sequences were queried against the BOLD Systems (https://www.boldsystems.org/) and GenBank (https://blast.ncbi.nlm.nih.gov/) for confirmation of species identification on February 23, 2022.

Table 1 Collection data for 12 African Saturniidae (Lepidoptera) specimens used for mitogenome sequencing, comparative mitogenomics and phylogenetic reconstruction.

Species	Tribe	Specimen	Life stage	Collection date	Country	Region	GPS		Edible	GenBank	
Bunaea alcinoe (Stoll, 1780)	Bunaeini	BA1.6	Adult	2019/10/11	South Africa	Gauteng	−25.7479	28.2293	Yes	OL912807	
Eochroa trimenii (C. & R. Felder, 1874)	Eochroini	ET6	Larva	2020/08/01	South Africa	Western Cape	−34.0771	18.4951	Not reported	OL912808	
Epiphora bauhiniae (Guérin-Méneville, 1832)	Attacini	GM5.2	Larva	2020/02/26	Namibia	Okonjima	−20.7579	16.7898	Yes	OL912809	
Gonimbrasia tyrrhea (Cramer, 1776)	Bunaeini	GT1.1	Adult	2020/05/25	South Africa	Western Cape	−32.7681	18.1607	Yes	OL912810	
Heniocha apollonia (Cramer, 1779)	Bunaeini	HA1	Adult	2020/10/04	South Africa	Western Cape	−34.0367	23.0495	Yes	OL912811	
Heniocha dyops (Maassen, 1872)	Bunaeini	GM8.2	Adult	2020/03/15	Namibia	Windhoek	−22.8577	17.1089	Yes	OL912812	
Holocerina smilax (Westwood, 1849)	Micragonini	HS5	Larva	2020/01/25	South Africa	KwaZulu- Natal	−26.15	27.9637	Not reported	OL912813	
Ludia delegorguei (Boisduval, 1847)	Micragonini	LD1	Larva	2020/12/12	South Africa	KwaZulu- Natal	−29.8455	30.9197	Not reported	OL912814	
Nudaurelia cytherea (Fabricius, 1775)	Bunaeini	NC1.2	Larva	2020/10/12	South Africa	Western Cape	−34.3322	18.9878	Yes	OL912953	
Nudaurelia wahlbergii (Boisduval, 1847)	Bunaeini	NW1	Larva	2020/12/11	South Africa	KwaZulu- Natal	−29.8244	30.9263	Yes	OL912817	
Vegetia ducalis (Jordan, 1922)	Micragonini	VD1	Adult	2020/05/22	South Africa	Western Cape	−33.4201	18.4005	Not reported	OL912815	
Vegetia grimmia (Geyer, 1832)	Micragonini	VG2	Adult	2020/03/12	South Africa	Western Cape	−34.3053	18.4619	Not reported	OL912816	

Sequencing, assembly, and annotation of mitogenomes

Specimens were individually sequenced using the Ion Torrent™ S5™ platform (ThermoFisher Scientific, Waltham, MA, USA) available at the Central Analytical Facilities of Stellenbosch University, South Africa. Sequence libraries were prepared using the Ion Xpress™ Plus gDNA Fragment Library Kit (ThermoFisher Scientific, Waltham, MA, USA), according to the protocol MAN0009847, REV J.0. Libraries were pooled and sequenced using the Ion 540™ Chef Kit (ThermoFisher Scientific, Waltham, MA, USA). The Ion Torrent reads were quality trimmed with a 30-base sliding-window at an average threshold of Q16. The remaining sequencing reads were then filtered for read length, where any reads shorter than 25 bases were removed from the data. Reference-based assembly of the next-generation sequencing (NGS) reads was performed using mitogenomes of the closest Saturniidae relatives available on GenBank (Table S1). NGS reads were assembled and mapped to the reference sequences on Geneious Prime v2021.1.1 (https://www.geneious.com/) using the “Map to reference” function, under the low to medium sensitivity option and fine-tuning up to five iterations. The open reading frames of the expected 13 mitochondrial protein-coding genes (PCGs) were identified with Geneious Prime using the invertebrate mitochondrial genetic code. Transfer RNA (tRNA) genes were located with ARWEN software (http://130.235.244.92/ARWEN/) (Laslett & Canbäck, 2008), under the default composite metazoan mitochondrial genetic code. The two ribosomal RNA (rRNA) genes and the non-coding AT-rich region were annotated by manual comparison to other Saturniidae mitogenomes available on GenBank. The raw sequence data were deposited on GenBank (PRJNA796275), as well as the 12 assembled and annotated mitogenomes (OL912807 to OL912817/ and OL912953).

Mitogenomic and phylogenetic analyses

Mitogenome nucleotide composition and biases [(AT-skew = (A−T)/(A+T) and GC-skew = (G−C)/(G + C)] were calculated in Geneious Prime. Gene overlapping regions and intergenic spaces were counted manually. Non-synonymous (Ka) and synonymous (Ks) substitution rates in PCGs were calculated in DnaSP6 (Rozas et al., 2017), and relative synonymous codon usage was calculated in MEGA X (Kumar et al., 2018), with the invertebrate mitochondrial genetic code in both cases. The presence of tandem repeat elements in the AT-rich region was searched using the Repeat Finder plug-in available on Geneious Prime.

The phylogenetic positions of the 12 African Saturniidae species were inferred in the context of other sequences available on GenBank for the family as of October 2021 (Table S2). The final dataset (n = 44) included 35 species in 19 genera and five tribes, with Bombycidae (Bombyx mori, Bombyx mandarina and Rondotia menciana), Sphingidae (Manduca sexta and Sphinx morio) and non-Bombycoidea (Biston panterinaria, Phthonandria atrilineata, Protantigius superans and Spindasis takanonis) as outgroups. Phylogenetic analyses were performed using the 13 PCGs (all codon positions). Individual PCG sequences were extracted from each mitogenome and stop codons were removed manually. Translation alignments were performed separately for each PCG using the MAFFT algorithm in Geneious Prime and then concatenated in a single alignment for each specimen. Poorly aligned regions and alignment gaps were removed using GBlocks v0.91b (Castresana, 2000). Two Bayesian inference (BI) and one maximum likelihood (ML) methods were used for phylogenetic reconstruction. The best partitioning scheme and model for MrBayes v3.1.2 (Huelsenbeck & Ronquist, 2001) and ML were determined using the edge-linked greedy strategy (Lanfear et al., 2012) in PartitionFinder v2.1.1 (Lanfear et al., 2017) built-in Anaconda programme (https://repo.anaconda.com/), and GTR+I+G was considered the best model. BI in MrBayes was performed under the GTR+I+G nucleotide substitution model selected with PartitionFinder. Analyses were performed as follows: two independent runs of four heat chains, ten million generations run simultaneously, resampling every 1000 generations. The first 25% of trees were discarded as burn-in, and the decision criterion for the convergence of the two runs was set as an average split frequency of ≤ 0.01. BI analyses were also performed under the site-heterogeneous mixture model CAT+GTR in PhyloBayes MPI in XSEDE v1.8c (Lartillot et al., 2013) to minimize the effect of mitochondrial compositional heterogeneity on phylogenetic reconstructions (Lartillot, Brinkmann & Philippe, 2007; Cai et al., 2020). Constant sites were removed from the alignment, and the minimum number of cycles was set to 30,000 with the burn-in set to 1000. The “maxdiff” was set to 0.3, and the minimum effective size was set to 50. Nodal support in the MrBayes and PhyloBayes trees was estimated as Bayesian posterior probabilities (BPP). ML analyses were performed in IQ-TREE v1.6.12 (Nguyen et al., 2015) under the GTR+I+G substitution model selected by PartitionFinder. Branch supports were determined using 1000 replicates for both the ultrafast bootstrapping (UFBoot) and the SH-aLRT branch test (Guindon et al., 2010; Hoang et al., 2018). MrBayes, PhyloBayes and IQ-TREE analyses were run on the CIPRES Science Gateway Portal (Miller, Pfeiffer & Schwartz, 2010). The final trees were drawn using FigTree v1.4.4 (https://bio.tools/FigTree).

Results and Discussion

Mitochondrial phylogeny estimates of Saturniidae have been largely limited to Asian species, and the only mitogenomes publicly available for African species prior to this study were those of G. belina and Gy. maja, both in the tribe Bunaeini (Langley et al., 2020). This work adds the mitogenomes for 12 species of the tribes Bunaeini, Micragonini, Eochroini and Attacini and significantly expands the genetic resources available for African Saturniidae, thus allowing further insights into the mitochondrial phylogeny of the family.

DNA barcoding

The sequence queries against BOLD Systems confirmed the morphological identification for all specimens used for mitogenome sequencing with sequence similarity between 99.44 and 100% (Table S3). The BLASTn queries against GenBank produced no matches with high identity (>98%) except for B. alcinoe and N. cytherea due to the absence of sequences for these species on the database.

Comparative mitogenomics of Saturniidae

The Ion Torrent runs generated between 11,321,643 (V. ducalis) and 21,284,325 (E. bauhiniae) reads, with an average size of 184 bp. Total mapped reads ranged from 16,637 (H. smilax) to 191,295 (H. dyops). Average sequence coverage varied between 197× (H. smilax) and 1810× (H. dyops) (Table 2). The total length of the new mitogenomes ranged from 15,218 bp (V. ducalis) to 15,363 bp (N. cytherea), in line with those of other Saturniidae (Kim et al., 2017a; Kim et al., 2017b; Langley et al., 2020; Liu et al., 2021; Chen et al., 2021). All mitogenomes generated in this study are identical in gene content and organization and include the 13 PCGs, two rRNA genes and 22 tRNA genes typical of Metazoa. Nine PCGs (ATP6, ATP8, CYTB, COI, COII, COIII, ND2, ND3 and ND6) and 14 tRNAs are located on the major (J) strand, and four PCGs (ND1, ND4, ND4L and ND5), eight tRNAs and the two rRNAs are located on the minor (N) strand (Table S4). Gene order is conserved across all species and identical to that of other Saturniidae, with the reciprocal translocation of tRNAMet and tRNAGln relative to tRNAIle (M-I-Q) as the only difference to the hypothetical ancestral arrangement for the non-Ditrysia lineage Hepialoidea of Lepidoptera (I-Q-M) (Cao et al., 2012) (Fig. 2).

All tRNAs have the typical cloverleaf structure except for tRNASer1 in B. alcinoe, E. bauhiniae, G. belina, G. tyrrhea, G. maja, H. apollonia, H. dyops, H. smilax, L. delegorguei, V. ducalis and V. grimmia, in that the dihydrouridine (DHU) arm is absent, as occurs in many Metazoa (Cameron, 2014) (Fig. 3). In contrast, the DHU arm of tRNASer1 is present in E. trimenii, N. cytherea and N. wahlbergii. The length of the tRNAs ranges from 57 bp (tRNAPhe) in N. wahlbergii to 74 bp (tRNATrp) in H. apollonia. The location and average size of 16S rRNA (1,357 bp; between tRNALeu1 and tRNAV al) and 12S rRNA (760 bp; be-tween tRNAV al and AT-rich region) across the 12 species is in line with the average size and position of the two genes in other Saturniidae. No tandem repeat elements were detected in the AT-rich regions of the 12 species.

Table 2 Results of next-generation sequencing for the assembly and annotation of the complete mitochondrial genomes of 12 African Saturniidae (Lepidoptera) species.

Specimen	Number of reads	Average read length (bp)	Number of rmapped reads	Average coverage	Sequence length (bp)	
Bunaea alcinoe BA1.6	15,634,521	187	49,550	582×	15,305	
Eochroa trimenii ET6	16,515,662	182	62,243	718×	15,254	
Epiphora bauhiniae GM5.2	21,284,25	173	108,486	1147×	15,259	
Gonimbrasia tyrrhea GT1.1	17,135,076	177	132,321	1412×	15,296	
Heniocha apollonia HA1	16,037,132	186	137,494	1397×	15,270	
Heniocha dyops GM8.2	15,407,906	173	191,295	1810×	15,306	
Holocerina smilax HS5	19,181,464	192	16,637	197×	15,220	
Ludia delegorguei LD1	14,679,181	196	26,826	320×	15,238	
Nudaurelia cytherea GM5.2	19,034,516	174	33,793	349×	15,363	
Nudaurelia wahlbergi NW1	13,837,894	188	30,529	333×	15,287	
Vegetia ducalis VD1	11,321,643	195	42,382	509×	15,217	
Vegetia grimmia VG2	14 153 225	193	88 880	1048×	15,251	

Figure 2 Mitochondrial gene organization.

Linear map of the complete mitochondrial genomes of (A) African Saturniidae and (B) hypothetical ancestral of the non-Ditrysian lineage Hepialoidea of Lepidoptera.

Figure 3 Structure of tRNA-Ser in 12 African Saturniidae species.

Predicted structure of tRNASer1 in the complete mitochondrial genomes of 12 new and two previously sequenced African Saturniidae species. Inferred canonical Watson-Crick bonds are represented by lines, and non-canonical bonds are represented by dots.

The new sequences have the high AT content characteristic of insect mitogenomes, ranging from 78.80% in H. apollonia to 81.00% in E. trimenii (Table S5). The AT content of the total PCGs follows the same trend and varies from 77.40% in H. apollonia to 79.90% in E. trimenii. Average AT content across the 12 species is highest in ATP8 (92.70%) and lowest in COI (69.30%) (Fig. 4), in line with previous reports on Saturniidae (Jiang et al., 2009; Singh et al., 2017; Chen et al., 2021). AT-skew is negative for PCGs on the J-strand across all species and varies from −0.22 in ND3 to −0.02 in ATP8. All four PCGs on the N-strand have a positive AT-skew, ranging from 0.14 in ND5 to 0.24 in ND1 and ND4L. Additionally, most PCGs have a slightly negative GC-skew, ranging from −0.69 (ATP8) to −0.02 (COI), except for COIII = 0.00 in L. delegorguei.

Figure 4 Nucleotide composition of Saturniidae mitogenomes.

Nucleotide composition in 14 complete mitochondrial genomes of African Saturniidae (Lepidoptera). (A) AT% content, (B) GC% content, (C) AT-skew, and (D) GC-skew for individual genes, total protein-coding genes (PCG), individual rRNAs and complete mitogenomes.

All 13 PCGs initiate with standard ATN start codons except COI and COII, which start with the alternative codons CGA and GTG, respectively. CGA as the start codon for COI has been reported to be highly conserved in Lepidoptera (Margam et al., 2011), and GTG was reported as the start codon for COII in A. assamensis, E. pyretorum, S. boisduvalii and S. ricini (Jiang et al., 2009; Kim et al., 2012; Singh et al., 2017; Kim et al., 2017a; Kim et al., 2017b). PCGs terminate with complete TAA or TAG codons or with incomplete TA- or T-, which are presumed to be completed by posttranscriptional modifications such as polyadenylation (Salinas-Giegé, Giegé & Giegé, 2015) (Fig. 5, Table S6). The most frequent amino-acid codons are Leu, Ile, Phe and Ser across all species, whereas Cys, Asp, Arg, Glu and Gln are rare (Fig. 6A). The most frequently used codons are AT-rich, and this feature is reflected in the total mitogenome nucleotide bias towards A and T. ATP8 has the largest number of different codons compared to all other genes across the 12 species (Fig. 6B). Relative synonymous codon usage (RSCU) is higher than 1.0 for all codons and highest for Leu1 (Fig. 7). Average Ka/Ks was found to be less than 1.0 in all PCGs across all species, indicating purifying or stabilizing selection, and to be highest for ATP8 (Ka/Ks = 0.31) (Fig. 8).

Figure 5 Usage of start and stop codons in Saturniidae mitogenomes.

Usage of start and stop codons of 13 protein-coding genes in 14 mitogenomes of African Saturniidae species.

Figure 6 Codon frequencies in Saturniidae mitogenomes.

(A) Mean codon frequencies and (B) standard deviation in the mitochondrial protein-coding genes of 14 African Saturniidae. Vertical axis: Protein-coding genes; Horizontal axis: Single-letter amino acid codons.

Figure 7 Relative synonymous codon usage in Saturniidae mitogenomes.

Average relative synonymous codon (RSCU) usage in the mitochondrial protein-coding genes of 14 African Saturniidae species. Vertical axis: RSCU; Horizontal axis: Single-letter amino acid codons. Black dots represent outliers that differ significantly from other observations.

Figure 8 Ka, Ks and Ka/Ks rates in Saturniidae mitogenomes.

Rates of non-synonymous and synonymous substitutions in the 13 mitochondrial protein-coding genes of 14 African Saturniidae species. Vertical axis: protein-coding genes; horizontal axis: Ka - number of nonsynonymous substitutions per non-synonymous site; Ks -number of synonymous substitutions per synonymous site.

Phylogenetic position of African Saturniidae in the family

The phylogeny of the family Saturniidae has been reconstructed using morphological, nuclear and mitochondrial sequence data. Regier et al. (2008) analysed four protein-coding nuclear genes using parsimony and ML methods and included five tribes present in Africa (Attacini, Bunaeini, Micragonini, Saturniini and Urotini). The recovered tribal structure strongly supported Attacini as sister group of Saturniini, which together formed the sister group of Urotini, Bunaeini and Micragonini, although with Urotini paraphyletic in regard to Bunaeini and Micragonini. More recently, the phylogeny of Saturniidae was reconstructed using a phylomorphology approach based on anchored-hybrid-enriched (AHE) loci followed by geometric morphometrics of hindwings (Rubin et al., 2018). The tree also supported a sister relationship between Attacini and Saturniini, the African tribes Bunaeini, Urotini and Micragonini together forming a monophylum and the sister group of Attacini+Saturniini, with Urotini again paraphyletic in regard to Bunaeini. In addition to phylogeny estimates based on nuclear genes and phylomorphology, the relationships among Saturniidae have been reconstructed using mitogenomic data (Sima et al., 2013; Kim et al., 2018; He, Wang & Chen, 2017; Zhong et al., 2017; Langley et al., 2020; Chen et al., 2021; Liu et al., 2020; Liu et al., 2021; Zhao et al., 2021). Although these studies focused on Asian species in the tribes Attacini and Saturniini, reciprocal monophyly and sister relationships were largely consistent, except for the positions of Cricula trifenestrata and Neoris haraldi, which were not consistently recovered, likely due to low sequence quality. The addition of the African tribe Bunaeini to the mitochondrial phylogeny did not challenge the sister relationship between Attacini and Saturniini, which together formed the sister group of Bunaeini (Langley et al., 2020; Chen et al., 2021), in agreement with the non-mitochondrial phylogeny estimates (Regier et al., 2008; Rubin et al., 2018). Thus, mitochondrial phylogeny estimates of Saturniidae have included only the tribes Attacini, Bunaeini and Saturniini, and African species were severely under-represented compared to Asian species (Langley et al., 2020; Chen et al., 2021). To contribute to the filling of this gap, we assessed the phylogenetic position of 12 African species of Saturniidae including the first representatives of the tribes Eochroini and Micragonini. Due to the suspected low quality of the mitogenomes of C. trifenestrata and N. haraldi, we performed phylogeny estimates with and without these sequences.

Phylogenetic trees excluding C. trifenestrata and N. haraldi

The ML and MrBayes trees excluding C. trifenestrata and N. haraldi (Fig. 9) displayed the same phylogenetic structure, recovering Bunaeini as a basal lineage, a sister relationship between Micragonini and Eochroini and the sister relationship between Saturniini and Attacini as found in previous analyses based on nuclear genes (Regier et al., 2008), phylomorphology (Rubin et al., 2018) and mitogenome data (Langley et al., 2020; Chen et al., 2021) (Fig. 9). In contrast, the PhyloBayes tree (Fig. 10) recovered a topology different from those of ML and MrBayes and of previous phylogenetic hypotheses in that Saturniini appear non-monophyletic, and several nodes have lower statistical support. Therefore, our discussion will focus on the topology recovered by the ML and MrBayes methods.

Figure 9 ML and MrBayes trees of Saturniidae species.

Maximum likelihood and MrBayes trees of the mitochondrial phylogeny of the family Saturniidae (Lepidoptera) based on 13 protein-coding genes (all codon positions). (A) MrBayes tree under the GTR+G+I model. Nodal support is given as Bayesian posterior probability. (B) Maximum-likelihood tree. Nodal support is shown as ultrafast bootstrap support (UFBoot)/approximate likelihood ratio test (SH-aLRT).

Figure 10 PhyloBayes tree of Saturniidae species.

PhyloBayes tree of the mitochondrial phylogeny of the family Saturniidae (Lepidoptera) based on 13 protein-coding genes (all codon positions) under the CAT+GTR mixed model. Nodal support is given as Bayesian posterior probability.

Eochroa trimenii, traditionally classified in Bunaeini (e.g., Bouvier, 1936; Pinhey, 1972; Staude et al., 2016; Staude et al., 2020), was not recovered in this tribe but as sister group of Micragonini, albeit with full nodal support only in the MrBayes tree. Its exclusion from Bunaeini had earlier been proposed by Oberprieler (1997) and was formalized by Cooper (2002), who placed it in a separate tribe, Eochroini. This result cannot be compared with previous mitochondrial and non-mitochondrial phylogeny estimates, because Eochroa is here included in a phylogenetic analysis for the first time. Although our result does not discount the possible inclusion of Eochroa in Micragonini, such a placement is repudiated by the fact that Eochroa does not possess the synapomorphic characters of Micragonini, as outlined by Oberprieler & Nässig (1994) and Oberprieler (1997). Furthermore, its relationship to the genus Usta, which appears in a similar position as sister group of Micragonini in the analyses of Regier et al. (2008) and Rubin et al. (2018), needs to be investigated. The ML and MrBayes trees recovered the monophyly of the tribes Bunaeini and Micragonini with full nodal support, and the results were broadly in agreement with previous phylogeny estimates (Rubin et al., 2018; Chen et al., 2021).

ML and MrBayes outperformed PhyloBayes in that the broad tribal structure of the first two methods is congruent with those of non-mitochondrial phylogeny estimates (Regier et al., 2008; Barber et al., 2015; Rubin et al., 2018), except that the tribes Micragonini and Eochroini are not recovered as sister group of Bunaeini but of Attacini+Saturniini instead. Different phylogenetic reconstruction methods (i.e., Bayesian inference vs ML) may impact tree topology, especially with regards to the order of deeper nodes. For example, a study of the high-level phylogeny of the Coleoptera inferred with mitogenome sequences showed that different inference models (ML and Bayesian inference) can yield inconsistent topologies for the same data (Yuan et al., 2016). Additionally, our analyses did not include all African Saturniidae lineages (missing in particular the various genera currently classified in Urotini), and this incomplete sampling may also not be sufficient for a well-supported recovery of the tribal structure of the family. For example, mitogenomic phylogeny estimates of Lepidoptera (Timmermans, Lees & Simonsen, 2014) and Hymenoptera (Mao, Gibson & Dowton, 2015) have failed to resolve some of the relationships with confidence, likely due to limited taxon sampling. The inclusion of representatives of Urotini will allow further insights into the mitochondrial phylogenetic reconstruction of Saturniidae, as this tribe was non-monophyletic in the phylogeny estimates based on nuclear genes (Regier et al., 2008) and phylomorphology (Rubin et al., 2018).

Phylogenetic trees including C. trifenestrata and N. haraldi

The phylogenetic analyses including C. trifenestrata and N. haraldi (Figs. S1–S2) resulted in lower support for the monophyly of Saturniini, and N. haraldi was recovered in a basal position relative to all other Saturniidae. The genus Neoris was not represented in the phylogeny estimates derived from nuclear genes (Regier et al., 2008) or phylomorphology (Rubin et al., 2018); therefore, it is not possible to establish comparisons. The monophyly of Attacini had low support in the ML tree with Rhodinia fugax, traditionally classified in Saturniini but morphologically similar to Attacini, as the basal-most member of Attacini, but the Bayesian trees more strongly and consistently supported the monophyly of this tribe including R. fugax. Cricula was recovered as sister group of Antheraea in non-mitochondrial phylogeny estimates (Regier et al., 2008; Rubin et al., 2018), but the position of C. trifenestrata was recovered inconsistently in our trees as well as in a previous work, where it formed a group diverged from the other Saturniini along with N. haraldi (Langley et al., 2020). In the present study, we detected several issues in the mitogenomes of C. trifenestrata and N. haraldi, including cases of shifts in the reading frame caused by artefactual single nucleotide indels that we corrected, but it is possible that other sequencing errors evaded our curation. Therefore, the positions of N. haraldi and C. trifenestrata seem be the result of substandard sequencing quality of these publicly available mitogenomes.

Conclusions

Prior to this study, baseline mitogenomic data for African Saturniidae with utility for assessments of the genetic diversity, phylogeography and phylogenetic relationships in the family were scarce compared with Asian counterparts. We sequenced the mitogenomes of 12 African Saturniidae species, including the first representatives of the tribes Eochroini and Micragonini, and significantly expanded the available genetic information for this group, which contains numerous species of economic, nutritional and cultural importance in sub-Saharan Africa. Our results support the monophyly of the tribes Bunaeini, Micragonini, Saturniini and Attacini, but with E. trimenii excluded from Bunaeini and placed in a separate tribe, Eochroini, and as sister group of Micragonini, and the sister relationship between Saturniini and Attacini, with the latter tribe including R. fugax as its basal-most member. However, a sister group relationship between Micragonini+Eochroini and Bunaeini was not consistently found, recovered only in the PhyloBayes reconstruction but not in the ML and MrBayes ones. These present findings contribute towards a more comprehensive understanding of the diversity and relationships among African Saturniidae.

Supplemental Information

Supplemental Information 1 Species and references

Species sequenced in this study and reference sequences used for the mapping of NGS reads of each species for recovery of complete mitogenomes.

Click here for additional data file.

Supplemental Information 2 Mitogenomes used in phylogenetics

Complete mitogenome sequences of 35 species used in the phylogenetic reconstruction of the family Saturniidae (Lepidoptera), including new and publicly available data as of 9 August 2021. n. a. –information not available.

Click here for additional data file.

Supplemental Information 3 DNA barcode queries

Results of BOLD and BLASTn queries of DNA barcodes of 12 species of African Saturniidae.

Click here for additional data file.

Supplemental Information 4 Mitogenome features

Main features of the complete mitogenomes of 12 African Saturniidae (Lepidoptera) species. N - minority strand; J –majority strand; IGN –number of intergenic nucleotides (negative values indicate overlapping genes).

Click here for additional data file.

Supplemental Information 5 Nucleotide composition and bias

Nucleotide composition, AT and GC-skewness in complete mitogenomes, combined protein-coding genes, combined rRNAs, individual genes, and individual rRNAs of 12 new and two previously published African Saturniidae (Lepidoptera) species.

Click here for additional data file.

Supplemental Information 6 Start and stop codons

Usage of start and stop codons in the mitogenomes of 12 new and two previously published African Saturniidae (Lepidoptera) species.

Click here for additional data file.

Supplemental Information 7 Bayesian trees

Bayesian inference trees of Saturniidae species based on 13 mitochondrial protein-coding genes. Nodal support is given as Bayesian posterior probabilities.

Click here for additional data file.

Supplemental Information 8 ML tree

Maximum likelihood tree of Saturniidae species based on 13 mitochondrial protein-coding genes. Nodal support was based on 1,000 bootstrap replicates.

Click here for additional data file.

The authors are grateful to Bronwyn Egan (University of Limpopo), Gail Morland (Namibia University of Science and Technology), William Versfeld (Ongava Research Center), Chrissie Fourie and Margriet Brink for assistance with specimen collection.

Additional Information and Declarations

Competing Interests

Author Contributions

Field Study Permissions

Data Availability

The authors declare there are no competing interests.

Zwannda Nethavhani, Rieze Straeuli and Kayleigh Hiscock performed the experiments, analyzed the data, prepared figures and/or tables, authored or reviewed drafts of the paper, and approved the final draft.

Ruan Veldtman conceived and designed the experiments, authored or reviewed drafts of the paper, and approved the final draft.

Andrew Morton conceived and designed the experiments, prepared figures and/or tables, authored or reviewed drafts of the paper, and approved the final draft.

Rolf G. Oberprieler and Barbara van Asch conceived and designed the experiments, authored or reviewed drafts of the paper, and approved the final draft.

The following information was supplied relating to field study approvals (i.e., approving body and any reference numbers):

Specimen collections were approved by Ezemvelo KZN Permits Office (OP408/2020), Cape Nature (CN44-59-13497) and National Commission on Research Science and Technology (AN20190911).

The following information was supplied regarding data availability:

The raw Next-Generation Sequencing data is available at GenBank: PRJNA796275. The mitogenomes reported are available at GenBank: OL912807 to OL912817 and OL912953.

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
