# Peer review of "Mitogenomics and phylogenetics of twelve species of African Saturniidae (Lepidoptera)"

_PeerJ, doi:10.7717/peerj.13275_

## Round 0.1 · original submission · Major Revisions

Dear Dr. Nethavhani and colleagues:

Thanks for submitting your manuscript to PeerJ. I have now received two independent reviews of your work, and as you will see, the reviewers raised some concerns about the research. Despite this, these reviewers are optimistic about your work and the potential impact it will have on research studying saturniid systematics and evolution. Thus, I encourage you to revise your manuscript, accordingly, taking into account all of the concerns raised by both reviewers.

While the concerns of the reviewers are relatively minor, this is a major revision to ensure that the original reviewers have a chance to evaluate your responses to their concerns. There are many suggestions, which I am sure will greatly improve your manuscript once addressed.

Please ensure to make all aspects of your research reproducible; specifically, please clarify the points raised by the issues that need more explanation. Also, ensure that barcodes are submitted to a public repository.

Therefore, I am recommending that you revise your manuscript, accordingly, taking into account all of the issues raised by the reviewers.

Good luck with your revision,

-joe

·

Basic reporting

The family Saturniidae is importantly economic insects used for silk-producing or edible species. However, information regarding African Saturniidae is very limited. This study provides 12 new complete mitochondrial genomes of African Saturniidae species and offers a well-resovled phylogenetic relationships in the family. The introduction into these species and why they are interesting is done well. The papers is well written the data seems to support the conclusions. Before acceptance for publication, some points should be addressed.
1. I suggest that the authors can provide the accession nos. (in GenBank) of 12 new mitochondrial genomes in Table 1.
2. I commend the authors to perform a DNA barcode analysis for 12 Saturniidae species with the known vouchers in BOLD system (www.BOLDsystem.org) and GenBank to ensure that the samplings are right. By searching against BOLDsystem and GenBank, only two species, Nudaurelia wahlbergii and Vegetia grimmia, have not available DNA barcode sequences. Thus, I ask authors to please submit the DNA barcode of the voucher used for the mitogenome to BOLD, along with its metadata (collecting data, photo,….).
3. If possible, please perform a search for tandem repeat element in the A + T-rich region, as found Antheraea pernyi and Antheraea roylei.
4. Line 28: Bayesian = Bayesian inference.
5. Line 66. mitochondrial genome = mitochondrial genome (mitogenome). Please check throughout the text.
6. Line 82. Please provide a reference for the number of 24%.
7. Line 108. What is the size of the NGS reads?
8. Line 181. The ancestor type for non-Ditrysian lineage Hepialoide of Lepidoptera has been reprted by Cao et al. (2012) [BMC Genomics 13, 276].
9. Line 219. maximum likeihood = ML.
10, Some references are repeated.

Experimental design

no comment

Validity of the findings

no comment

Additional comments

no comment

Reviewer 2 ·

Basic reporting

The model name GTR+G+I is generally not preferred, please change it to GTR+I+G.
Please mention the accession number PRJNA796275 in the main manuscript.

Experimental design

The authors find largely consistent phylogenetic relationships among BI and ML analyses. The problems happened in Saturniini especially for the positions of Cricula trifenestrata and Neoris haraldi. They attribute the inconsistency to low quality of the mitogenomes of the two species. It should be noted that the low quality of sequences not only influence the related species themselves but also disturb the whole phylogeny, since the total validate sites will be reduced with erroneous alignments. If it is the case as the authors mentioned, it is worthy to try to rebuild trees without the two species.
Line 105. Do you filter the low quality reads?
Line 162. Please mention how you check and guarantee the completeness of the mitogenomes.

Validity of the findings

no comment

Additional comments

no comment

Reviewer 3 ·

Basic reporting

no comment

Experimental design

no comment

Validity of the findings

no comment

Additional comments

First of all, I think Authors rational to perform this study, experimental design and its execution are decently reflected in the manuscript. Nethavhani and colleagues sequenced the complete mitochondrial genomes of 12 species found in Southern Africa for comparative mitogenomics and phylogenetic reconstruction of the family. These results support the monophyly of the tribes Attacini, Bunaeini and Micragonini and the placement of E. trimenii and R. fugax in the tribes Eochroini and Attacini, respectively.
This will help to significantly expand the genetic data of Africa Saturniidae and thus provide a more comprehensive understanding of the diversity and relationships among African Saturniidae. Hence, it certainly deserves publication with some moderate changes as below:

1. The mitochondrial genome of Heniocha apolloniais has been determined, but why not put its picture in Figure 1?

2. Some of the statements in Table 1 and Figure 1 seem to be inconsistent.
In Table 1: Vegetia grimmia (Geyer, 1832);Nudaurelia wahlbergii (Boisduval, 1847);Heniocha dyops (Maassen, 1872) and Gonimbrasia tyrrhea (Cramer, 1775), but in Figure1: Vegetia grimmia (Geyer, 1831);Nudaurelia wahlbergii (Nudaurelia, 1972);Heniocha dyops (Maassen, 1886) and Gonimbrasia tyrrhea (Cramer, 1776).


3. Line 212-214,“Relative synonymous codon usage (RSCU) was higher than 1.0 for all codons and highest for Leu1 (Fig. 7). Average Ka/Ks was less than 1.0 in all PCGs across all species indicating purifying or stabilizing selection and was highest for ATP8 (Ka/Ks = 0.31) (Fig. 8). ” Does the position of Figure 7 and Figure 8 seem to be reversed? What do the black dots in Figure 8 mean? Consider whether it is necessary to annotate.

4. There are still some minor problems in the format of the full text, please check carefully again, for example, there are no blank lines in reference 395-401; Some species names are not italicized, such as Samia cynthia ricini in line 41, etc.

---

## Round 0.2 · accepted · Accept

Dear Dr. Nethavhani and colleagues:

Thanks for revising your manuscript based on the concerns raised by the reviewers. I now believe that your manuscript is suitable for publication. Congratulations! I look forward to seeing this work in print, and I anticipate it being an important resource for groups studying saturniid systematics and evolution. Thanks again for choosing PeerJ to publish such important work.

Best,

-joe

·

Basic reporting

The authors have addressed all of the comments.

Experimental design

no comment

Validity of the findings

no comment

Reviewer 2 ·

Basic reporting

Well written.

Experimental design

Well done.

Validity of the findings

Well done.

Additional comments

I am satisfied by the revision. All potential problems related to my previous comments / suggestions are appropriately resolved. I suggest accepting this paper.

Reviewer 3 ·

Basic reporting

no comment

Experimental design

no comment

Validity of the findings

no comment